## Introduction

 

**Subject Category:**
Biology (whole organism)

evolution

**Author for correspondence:**
Peter Woodford
e-mail: peterjwd@gmail.com

A contribution to the 'Inclusive Fitness' special collection.

# Evaluating inclusive fitness

## Peter Woodford

Union College, Schenectady, NY, USA

PW, 0000-0001-6827-644X

## 1. Background to this special collection

In 2010, a contentious debate erupted in the field of evolutionary biology in response to an article published in the journal *Nature* by two mathematical biologists, Martin Nowak and Corina Tarnita, and the renowned evolutionary theorist and entomologist, E.O. Wilson [1]. The article questioned the explanatory efficacy and value of William Hamilton's theory of 'inclusive fitness', the reigning theoretical and mathematical basis of decades of empirical research into the evolution of social behaviour—especially cooperative and altruistic behaviour—across the living world. It was an especially polarizing article, given that E.O. Wilson was once one of the main proponents of Hamilton's theory for explaining the evolution of sterile worker castes in insects, and in this article, he revoked his earlier stance. A number of highly critical responses followed, one signed by 137 eminent theoreticians and empiricists in evolutionary biology [2]. The number of scientists rejecting the conclusions of Nowak, Tarnita and Wilson was itself an indication of the nerve that it struck, and also of the continuing centrality of Hamilton's theory to the study of social evolution. But while the years since the publication of the article have witnessed a number of responses and counter-responses that have attempted to settle the disagreements, a definitive consensus is yet to emerge [3–9]. The first task of the present collection is to showcase diverse interpretations, evaluations and uses of inclusive fitness since then, and so to advance towards a better understanding of the role of this theory in diverse areas of research both in the life sciences and beyond. Our aim is to use the recent debates over inclusive fitness not only to highlight interesting divergences between uses of the theory, but also to explore the philosophical questions that it has raised about causality in the study of social evolution, and even about the nature of behaviour in general across diverse species.

## 2. The themed collection

This collection began as an interdisciplinary conference on disagreements over the interpretation and current status of Hamilton's work, and many of the papers have come from participants at that conference. We quickly found that the questions raised, by their nature, cut across a variety of disciplines and areas of specialization within the biological sciences, but also

in areas that draw on theoretical resources of the life sciences such as the emerging evolutionary social sciences, anthropology and philosophy. This interdisciplinary scope is thanks in large part to increasing advancement in applying theories of social evolution across the living world from cells to humans, and to more pressing questions about the generality of evolutionary principles. For this reason, this collection features articles from researchers in mathematical biology, behavioural ecology, anthropology and medicine to philosophy of science, and even ethical theory. It is thus premised on the belief that addressing all of the important questions raised by debates over how we explain social behaviour requires the input not only of various areas in biology, but also of philosophy and the social sciences.

Cooperation and altruism—and indeed social behaviour in general—are defined in evolutionary biology according to concepts of cost and benefit, in particular according to costs and benefits to the fitness of interacting organisms. The fitness effects of behaviours are apparent and measurable through interactions between actors and recipients. Altruistic behaviour, in particular, has been usefully defined as behaviour in which an actor pays a cost to its direct, lifetime net fitness and a recipient gains a benefit to its direct, lifetime net fitness [10]. Of course, important questions remain about how to define and measure costs and benefits to fitness, and whether or not these are best thought of as properties measurable in individual organisms, or rather at the level of populations or genes, but these concepts nonetheless define the nature of social behaviours and the puzzles they present [11,12]. The existence and maintenance of apparently costly forms of helping behaviour was of course something that Darwin puzzled over, and for a long time, it remained poorly understood until William Hamilton's recognition of the importance of *relatedness* between the actor and the recipient and his mathematical formalization of this insight in Hamilton's rule.

As was noticed by Hamilton, explaining biological altruism required re-examining some of the most basic concepts of evolutionary theory, most importantly the concept of individual fitness itself. The recent debates over Nowak *et al.*'s criticism have shown this as well. In debating Hamilton's rule, the concept of inclusive fitness, and the evolutionary process of kin-selection, the debate has also touched upon the most general aspects of evolutionary theory. As will be shown in the articles that follow, debating Hamilton's work involves reflecting on how we conceive of fitness and its measurement; how researchers approach the methodological interplay between theory and empirical observation; and how formal, mathematical tools orient empirical research and even shape our conception of evolution as a dynamic process. Moreover, thanks to the fact that these debates were sparked by the topic of altruism, they have once again brought to the foreground the question of what the evolutionary study of social behaviour can contribute to our explanation of the capacities and motivations evident in human behaviour, and in particular in the ethical and religious valuation of altruism.

## 3. Summary information

The articles in this collection together touch upon three key questions and are organized accordingly. The first question, as already stated, is simply how best to understand the key points of disagreement between defenders and critics of Hamilton's rule and the concept of 'inclusive fitness'.

To address this question, and act as something of a summary of the entire debate, we have Jonathan Birch's rigorous analysis in [13].

Philosophers Samir Okasha and Johannes Martens provide further analysis of an issue raised in debates over Hamilton's rule, namely, whether and how it gives insight into causes of social evolution. Their article [14] attempts to move the debate forward by exploring and responding to critics of inclusive fitness who argue that Hamilton's rule does not and cannot successfully describe causes of different patterns of social interaction.

The second question follows upon an adequate answer to the first, and this is how these debates help us understand the relationship between theoretical modelling and mathematical research and empirical work on real-world organisms in evolutionary biology. For example, do empiricists, theoreticians and mathematicians understand and use the explanatory tools of Hamilton's rule and the concept of inclusive fitness differently? Questions in this area are crucial, because advancing discussion over the place of Hamilton's work in the contemporary study of social evolution requires understanding the interplay between theoretical modelling and the observation and measurement of living systems—and vice-versa. To address this question, we have a number of specific studies that demonstrate assessments of kin-selection, inclusive fitness and Hamilton's rule. Each of these articles sheds light on how debate over Hamilton's theory has affected our understanding of the nature and extent of cooperative and altruistic behaviour across species and levels of biological organization.

Davies & Gardner [15] argue that Hamilton's insights into the importance of relatedness are upheld by evidence of the role of monogamy in societies in which costly helping evolves successfully.

Marta Bertolaso and Anna Maria Dieli's article considers the limitations of inclusive fitness and the need for a multi-level approach for understanding the evolutionary dynamics of cancer. Their article appears to challenge the generality of inclusive fitness in relation to systems that do not appear to be best thought of in its terms [16].

Dieter Lukas and Tim Clutton-Brock explore the role of climate, alongside typical explanations that invoke relatedness, in the evolution of cooperative breeding. Written by empiricists working on cooperative breeding, this article seeks to highlight that relatedness may be one of many other factors needed for costly helping to successfully evolve [17]. They thus challenge the primacy of relatedness as an ingredient in the evolution of costly forms of helping.

Cooney et al.'s [18] article explores a unique and challenging case of apparent altruism directed towards intruders and it draws lessons for what this means for biological interpretations of altruism.

Jussi Lehtonen and Lisa Schwartz analyse the equivalence of individual selection, kin-selection and group selection for models of sex ratio evolution. Their article also explores how theoreticians choose between different levels at which to describe evolutionary dynamics when they appear to yield identical results [19].

Josephine Brask et al. offer an example of non-kin cooperation outside of humans. While not an example of altruism, these findings may also support the criticism that costly forms of cooperation can evolve in the absence of relatedness [20].

Finally, we ask what, if any, implications these debates over inclusive fitness have for the explanation of human behaviour and for reflection on the nature of ethical values that encourage forms of altruism. To address this question, we bring together very different approaches to questions of ethics and human behaviour across philosophy, biology and the evolutionary social sciences.

Thomas et al.'s [21] article provides evidence in support of the value of inclusive fitness for understanding human behaviour.

Sibly & Curnow's [22] article offers a theoretical framework for understanding genetic contributions to altruistic behaviour that is based on Hamilton's rule, and they evaluate this framework in relation to empirical work.

Darragh Hare, Bernd Blossey and H. Kern Reeve attempt to explain the case of how altruistic regard for the welfare of other species might evolve—a case often thought to require abandoning biological principles of evolution and adopting theories of cultural evolution. Their article argues that inclusive fitness can be useful for questions in normative ethical theory about the moral status of species [23].

William Fitzpatrick argues against approaches represented by Hare, Blossey and Reeve. He argues that the applicability of evolutionary theory to the evaluation and justification of altruistic values is limited, and thus that there may also be limits on how evolutionary approaches can help us understand the nature of human altruism [24].

# 4. Interdisciplinary approach

A few more comments must be made about the interdisciplinary nature of what began, and might appear still, to be a strictly a local debate about worker sterility in insect societies. The reason that the recent debates over Hamilton's work have a wider resonance beyond evolutionary biology is the now quite mainstream realization that social dynamics and inter-dependencies are ubiquitous in the composition of the living world. Social insects are a microcosm of more general dynamics of cooperation and conflict that play themselves out at all levels of biological organization, in all lineages (including ours), and at all scales of evolutionary time. Principles of social evolution are poised to explain not only present interactions between observable organisms, but also so-called 'major transitions' by which the structures that now typically define whole organisms and their levels of organization first emerged. This feature of the constitution of biological systems and their evolution underlies the ways in which patterns of cooperation and competition are relevant across the academy, to philosophy and also the humanities [11]. From cells, to insects, to meerkats and humans, the development of the evolutionary science of sociality appears to provide a unifying umbrella for the study of what living things are, what they do and what wider implications the scientific understanding of life has for our own conception of ourselves, our humanity, and our social and moral pursuits.

The topic of these recent debates and the questions raised by them thus lend themselves to a multi-disciplinary approach. But another reason for the wider disciplinary circle that they touch upon is

provided by the fact that these debates raise questions that are philosophical. Let me say a few things to specify further what is meant by 'philosophical' here. By designating reflection as 'philosophical', the aim is not to claim disciplinary territoriality, hegemony or fundamentality—to place philosophy as the dominant 'queen' of the academic colony. Nor is it to suggest that the questions are unanswerable or merely interesting food for thought when one is sitting comfortably in an armchair. Instead, it is to claim the opposite, namely, that questions about what theories do for us, about their limitations and possibilities, are fundamental but not necessarily fully settled by particular studies. They often lie at the boundaries of what we currently know and understand. They thus lead into territory that is not ruled by any one discipline and that stretches the perspective of any particular specialized approach. Evaluation of Hamilton's explanation of altruism and its generality and wider significance is a problem like this. It requires areas of the sciences and specializations to come into contact and overlap, and to ask questions about fundamental concepts and methods. This meaning of 'philosophical' relates to the compatibility between different methods, concepts and questions. It emerges when we have to navigate between the study of specific systems towards general principles that capture a complex process like evolution and the complex, multi-level nature of its products. Finally, it relates to what emerges as work from particular specializations is stitched together into a 'bigger picture'.

There are further issues raised by these debates that are philosophical in another sense, and this is because they have centred on the topic of altruism. Even in Darwin's work, behaviours that appeared costly for actors and beneficial to recipients were of importance both for their general intelligibility within the dynamics of natural selection, but also because the topic of the evolution of altruism promised to provide naturalistic foundations for understanding the ethical motivations, ethical emotions, and normative ethical judgements that form the substance of human ethical life and practical reasoning. I use the term 'naturalistic' here in a loose sense to mean that these aspects of human psychology, cognition, and behaviour were made possible and put in place, so to speak, by the same processes that put in place the psychological and behavioural tendencies of other evolved animals. So, naturalistic here refers to the way in which human behaviour and human ethical life is to be made sense of through the same processes that pattern social behaviour in the living world more generally. The framework of theory in social evolution is the one that appears to make possible strong claims about the underlying unity between explanations of human social behaviour and the behaviour of other animals. This is why this collection ends with articles addressing human cooperation and ethics.

The recent debates over altruism and the papers here show that scientific interest in the possibility and extent of human altruism makes a difference for all sorts of 'extra-scientific' moral and political aims in which many do indeed have deep 'extra-scientific' interests, such as ones related to conservation that are touched upon in Darragh Hare *et al*. These moral and political issues are hardly in the foreground of the technical and formal questions at the heart of different evaluations of inclusive fitness, but they are part of the cultural context in which they are taking place and they are undeniably a part of why debates over altruism attract such interest and spark such controversy. Since the study of social evolution has played the role of the arch 'debunker' that has revealed altruism to be a disguised form of selfishness that can only evolve among relatives—where there is still genetic 'self-interest' at play—it has stood in awkward tension with extra-scientific goals that call for the extension of altruism to ever greater spheres. At the very least, biological research on altruism has shown that forms of costly helping can evolve, and it is probably a mere semantic issue to wonder whether or not these behaviours are 'really' selfish. Nonetheless, the legacy of Hamilton's work is tied to a conception of altruism that views it as an optimal individual adaptive strategy in certain social contexts, and thus to a conception of biological self-interest that is in some tension with our moral projects and self-understanding. Indeed, when pressing moral issues like those in debates about social justice, conservation and environmentalism ask us to push what biological theories seem to tell us about self-interest and about the limits of altruism, questions not only of the possibility of altruism in nature, but also of its value and meaning become even more pressing. This collection is one more step in continuing this valuable discussion of how, and whether, altruism is possible in nature.

Data accessibility. This article has no additional data.

Competing interests. I declare I have no competing interests.

Funding. The conference that led to this special collection was funded by the Templeton World Charity Foundation (TWCF).

Acknowledgements. This collection was made possible, in part, as a result of a conference held in May 2016 at the University of Cambridge. I thank Antonio Rodrigues in particular for help organizing that conference. I would also like to thank those who either participated in that conference or whose contributions were submitted specially for this collection: Jonathan Birch, Ruth Mace, William Fitzpatrick, Marta Bertolaso, Andy Gardner, Dieter Lukas and Tim Clutton-Brock.

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
