## [Reviewer comments · Royal Society Open Science]

Review History

Decision letter (RSOS-190644)

29-May-2019

Dear Dr Woodford:

It is a pleasure to accept your manuscript entitled "Introduction to "Evaluating Inclusive Fitness"" in its current form for publication in Royal Society Open Science. The comments of the Editor who reviewed your manuscript are included at the foot of this letter.

on behalf of Dr Claudia Wascher (Associate Editor) and Dr Kevin Padian (Subject Editor).

Associate Editor Dr Claudia Wascher Comments to Author:

Reports © 2019 The Reviewers; Decision Letters © 2019 The Reviewers and Editors; Responses © 2019 The Reviewers, Editors and Authors. Published by the Royal Society under the terms of the Creative Commons Attribution License <http://creativecommons.org/licenses/by/4.0/>, which permits unrestricted use, provided the original author and source are credited

Associate Editor

Comments to the Author:

The presented manuscript is an introduction into a special issue on the broad topic of inclusive fitness and the evolution of cooperation. The introduction starts with a historic overview over a recent debate over the inclusive fitness theory, which is very readable and gives an accessible and interesting introduction why a special issue in this area is highly relevant.

After this, the introduction give a bit of background over the origin of this special issue and gives a bit more background information why the debate is generally relevant in the field of evolutionary biology. Further, the author discusses the interdisciplinary relevance of the topic.

Overall the introduction is readable and gives a very nice overview over the topic.
